# Individual and Regional Characteristics Associated with Maternal Smoking During Pregnancy in Japan: Healthy Parents and Children 21

**DOI:** 10.3390/ijerph17010173

**Published:** 2019-12-25

**Authors:** Tadao Ooka, Yuka Akiyama, Ryoji Shinohara, Hiroshi Yokomichi, Zentaro Yamagata

**Affiliations:** Department of Health Sciences, University of Yamanashi, Yamanashi 409-3898, Japan; yukaa@yamanashi.ac.jp (Y.A.); rshinohara@yamanashi.ac.jp (R.S.); hyokomichi@yamanashi.ac.jp (H.Y.); zenymgt@yamanashi.ac.jp (Z.Y.)

**Keywords:** smoking, cigarettes, pregnancy, social capital, Healthy Parents and Children 21, Japan

## Abstract

Smoking during pregnancy causes various maternal and fetal health problems. Although there are considerable differences in maternal smoking proportions between localities, only a few studies have investigated the effects of regional characteristics on maternal smoking behavior. This study aimed to clarify the association between maternal smoking during pregnancy and individual and regional characteristics. We used data from a large nationwide birth cohort study in Japan that consisted of information on 20,267 women with children aged 3–4 months. The multilevel regression model was used to examine the association between smoking behavior during pregnancy and individual and regional characteristics. On multilevel analysis, late birth order, young age of the mother at birth, low birth weight, low economic status, husband’s smoking during pregnancy, maternal alcohol consumption during pregnancy, absence of a pregnancy counselor, and lack of participation in local events for childrearing were significantly associated with maternal smoking behavior during pregnancy at the individual level. Meanwhile, a high unemployment rate and a high number of nurseries were significantly associated with maternal smoking behavior during pregnancy at the regional level. In conclusion, we showed the relation between maternal smoking during pregnancy and the individual- and regional-level characteristics.

## 1. Introduction

Smoking during pregnancy has been suggested to cause various health problems to both the mother and the baby. It leads to low birth weight [1] and retarded intrauterine growth [1,2,3] and adversely affects the child’s growth after birth [4]. The recently proposed concept of Development Origins of Health and Disease suggests that child development and even their adulthood may be affected by their mothers’ smoking during pregnancy [5]. The World Health Organization has reported a recent increase in smoking rates among women and projected the average women’s smoking rate to increase to 20% by 2025 [6]. The prevalence of female smokers is the highest in the European and Americas regions. 

In Japan, the smoking rate of women has leveled at around 9% in the last 10 years (National Health and Nutrition Survey in Japan; 2014), and thus, new measures to reduce the smoking rate among women, particularly those who are pregnant, are needed. Though various programs against smoking during pregnancy have been implemented in Japan, the maternal smoking rate during pregnancy has not markedly declined in recent years. Furthermore, approximately 40% of women resume smoking after temporary cessation during pregnancy (from the survey and research project of the Health Promotion Association in Japan; 2013). 

Considering that there are significant differences in smoking rates by region, there is a considerable possibility of causing maternal smoking behavior by some characteristics of their living area. Therefore, the effect of an individual intervention for smoking cessation will be poor in the community with a high smoking rate because the environment of pregnant women does not encourage smoking cessation [7]. In addition, smoking pregnant women have difficulty trying to stop smoking because of several reasons including restrictions on medication use [8], and thus, community support is crucial for smoking cessation [9]. Further, countermeasures with a framework including social capital are also necessary [10].

Previous studies have shown some characteristics in pregnant women to influence smoking behavior; these include education levels [11], maternal parity [12], relationships with partners [13], and smoking among the partner or parents [14]. Conversely, a European study [10] found certain regional characteristics including regional socioeconomic status (SES) and ethnicity to be associated with maternal smoking behavior.

Smoking during pregnancy is considered to be an important factor in the association between low SES and negative infancy outcomes [15,16]. A low level of education and economics status, unhealthy behavior of family members, and poor relationships with close people predispose pregnant women to pursue habits that are inappropriate for childcare, including maternal smoking; this results in negative infancy outcomes. No studies have examined whether the regional SES plays an important role in Asia. It is therefore necessary to determine whether the same causal association exists in this area. It is also necessary to verify whether the influence of individual factors changes depending on regional factors, particularly those related to economic status. This study aimed to clarify the effects of regional factors on smoking during pregnancy. Towards this goal, we examined measures that might influence smoking behavior not only at the individual but also at the community level. Further, we examined both individual- and regional-level factors that might influence smoking behavior during pregnancy.

## 2. Materials and Methods 

### 2.1. Study Participants

This study used data from the final evaluation of Healthy Parents and Children 21. Healthy Parents and Children 21 was a national campaign of the Ministry of Health, Labor and Welfare of Japan aimed to improve the health status of mothers and children in Japan. The campaign started in April 2001 and ended in March 2015. A final evaluation of this campaign was performed in 2013. This evaluation was conducted by approximately 75,600 caregivers who implemented health checks for 3- or 4-month-old, 18-month-old, and 3-year-old children in 472 municipalities assigned for each city (Japan is divided into 47 prefectures. Each prefecture comprises numerous municipalities, with 1,719 in total.)

### 2.2. Data Collection

A total of 20,729 women with children aged 3 or 4 months old participated in this evaluation. We requested them to answer self-administered questionnaires sent by mail and gathered them when they came for a health check for their child. Of the 23,224 women who received the questionnaire, 20,729 women responded. Among them, 20,276 with available data were included in the analysis. The participant selection flow chart is shown in Figure 1.

The self-administered questionnaire comprised 40 questions that mainly evaluated the mothers’ parenting environment during pregnancy. All questions were answered with a yes or no or a numbered selection. 

### 2.3. Exposure Variables

We performed a single regression analysis for all questions to evaluate the influence of individual factors on pregnant women. There were 18 questions that were significantly related: “How many children do you have?”“How old were you at the time of your childbirth?”“How much did your child weigh at birth?”“How do you feel about your current economic status?”“Was your husband smoking during your pregnancy?”“Were you drinking during pregnancy?”“Were there any counsellors during and after your pregnancy?”“Have you participated in local events for child-raising?”“Do you want to have the next child?”“Did you have any unaddressed problems during your pregnancy or childbirth?”“Did you smoke when you found out you were pregnant?”“Was your husband smoking when you found out that you were pregnant?”“Did you feel the effects of the Japanese maternity mark during pregnancy?”“Does anyone speak to you on the road when you go out with your child?”“When you put your child to bed, do you make him/her lie on his/her back?“Do you breastfeed your child?”“Are you currently smoking?”“Is your husband currently smoking?”

To avoid obvious multicollinearity, questions regarding maternal and paternal smoking behavior were limited to the period of pregnancy (6 and 7). We excluded questions with too many options (15 options exist on question 10). We also excluded questions that would not have a direct effect on maternal smoking behavior (13–16), based on previous studies [7,9,10,11,12,13,14,17,18,19,20,21,22,23,24,25,26,27,28]. Finally, we selected 9 questions (1–9) as individual variables. We also implemented a stepwise analysis of these questions and chose eight questions as explanatory variables to regulate the individual-level model (See in Table 1).

For regional factors, we used 11 sociodemographic factors from the national database of the Japanese statistics bureau (e-stat, 2010; Statistics Bureau, Ministry of Internal Affairs and Communications, Japan). Single regression analysis of these factors identified five significant variables, namely, unemployment rate, birthrate, number of nurseries, financial strength index (FSI), and population density; and they were selected as regional factors.

### 2.4. Outcome Variable

The main outcome variable was smoking during pregnancy. This was evaluated using the question “Were you smoking during pregnancy?” and we determined the mothers’ smoking habit during pregnancy from the answer to this question. We evaluated the effects of individual factors on pregnant mothers and regional factors in their living area. The score of all regional factors was divided into quartiles (Q1–4). The definition of each individual and regional factor is described in Table 1.

### 2.5. Statistical Analysis

Multilevel logistic models were used to investigate the effect of regional factors on maternal smoking behavior during pregnancy. In this model, individual factors of pregnant women were used as the primary level, and regional factors of their living areas were used as the secondary level. Multilevel logistic regression analysis with random intercept models was employed to demonstrate the effect of regional factors on pregnant women. The estimated multilevel regression model is formally expressed as:

Individual-level model:Yij = β0j +β1 (Birth order) + β2 (Mother’s age) + β3 (Birth weight) +β4 (Economic status) + β5 (Husband’s smoking) + β6 (Drinking habit) + β7 (Existence of counselor) + β8 (Participation in local events) + eij
where Yij is the frequency of maternal smoking during pregnancy for the respondent i in the region j.

Regional-level direct effects model:β0j = γ00 + γ01 (Unemployment rate) + γ02 (Birthrate) + γ03 (Number of nursery) + γ04 (Financial Strength Index) + γ05 (Population density) + u0j

We use generalized linear mixed-effects models with adaptive Gauss Hermite quadrature. These models were fitted with the help of SAS PROC GLIMMIX. (The GLIMMIX procedure fits statistical models to data with correlations or nonconstant variability and where the response is not necessarily normally distributed.) All analyses were conducted using SAS version 9.4 (SAS Institute Inc., Cary, NC, USA).

### 2.6. Ethical Considerations

This study was approved by the Ethics Committee of the University of Yamanashi (identification code: No. 1119). The participants in Healthy Parents and Children 21 were informed that participation in this study was voluntary, and completion and return of the self-administered questionnaire indicated their consent to participate in this study. 

## 3. Results

Smoking was prevalent in more than 10% of women with very poor economic status (11.87%), four or more children (13.43%), and drinking habit during pregnancy (15.25%). Table 2 shows the prevalence of smoking women according to each individual- and regional-level variable. 

Several variables were significantly associated with maternal smoking during pregnancy. The results of single logistic regression analysis for each variable at the individual and regional level are shown in Table 3. Figure 2 shows the coefficient of correlation between each regional variable, and Figure A1 shows the coefficient of correlation between each individual variable.

In multilevel analyses (Table 4), Model 1 is an analysis of intercept, while Model 2 is a logistic regression analysis only with individual factors in consideration of regional level nesting. The characteristics of individual pregnant women that were significantly associated with smoking are birth of third child or more, age at birth of 24 years old or younger, infant birth weight of 2500 g or less, subjective economic status of poor or very poor, the husband’s smoking during pregnancy, maternal drinking habit during pregnancy, absence of counselor, and no participation in local events for childrearing. The most common group of each individual variable was selected as a reference for the analysis.

Model 3-1 to Model 3-5 are the results of the multilevel analysis with individual factors of pregnant women as the primary level and regional factors as the secondary level. The first quantile of each regional variable was selected for reference in the analysis. Unemployment rate was significantly associated with maternal smoking on Q3 (OR: 1.41, 95% confidence interval (CI): 1.102–1.805) and Q4 (OR: 1.477, 95% CI: 1.160–1.880). In Model 4, which is a multilevel analysis of all individual variables and all regional variables, the following were significantly associated with maternal smoking: birthrate on Q3 (OR: 0.638, 95% CI: 0.488–0.834), number of nursery on Q4 (OR: 1.406, 95% CI: 1.044–1.894), and unemployment rate on Q3 (OR: 1.439, 95% CI: 1.118–1.852) and Q4 (OR: 1.481, 95% CI: 1.147–1.914). As shown in Table A1, we conducted a stepwise analysis of regional variables and selected three as explanatory variables to regulate the sociological-level model. The number of nursery on Q4 was not significantly associated with maternal smoking in Model 4.

## 4. Discussion

In this study, we revealed the individual and the regional factors associated with maternal smoking by using individual data from a large national survey and regional data from the national statistics bureau. With respect to individual characteristics, all individual characteristics analyzed in this study were significantly associated with maternal smoking. With respect to regional factors, adjusted multilevel analyses revealed that unemployment rate, birthrate, and number of nurseries were significantly associated with maternal smoking behavior. These analyses indicated that maternal smoking behavior varies depending on regional characteristics.

The association of some individual factors on maternal smoking behavior has already been reported in previous studies [7,12,19,20,21,23,24]. Although most previous studies were conducted in Western countries, similar results in this Japanese study suggest that these factors are not affected by the living environment or ethnic differences. Smoking pregnant women were particularly prevalent in groups where the birth order of the expected child was at least the fourth, the mother was 19 years of age or younger, and the subjective economic situation was very poor. These results will serve to identify the group in particular need of smoking countermeasures in Japan. In addition, we newly showed that the existence of consultation partners and participation in local events for childrearing have a significant relation on their behavior. Both elements are related to loneliness, and it has been suggested in past systematic reviews that loneliness is related to smoking [29].

Regional factors in this study sufficiently adjusted our multilevel model with individual factors when we look at the small value of the random intercept of Model 4. Particularly, the unemployment rate was strongly related to the maternal smoking behavior. A Swedish study [10] has shown that regional economic conditions are related to maternal smoking during pregnancy. In this study, we considered unemployment rate and the Financial Strength Index (FSI) as regional factors related to economic condition, and only the unemployment rate was significantly associated with maternal smoking. The FSI indicates the wealth of the municipality itself, and the abundance of municipality does not mean the richness of the local residents, while high unemployment rates are apparently related to worse economic conditions.

The unemployment rate is also known to be related to the difficulty in smoking cessation [30], type 2 diabetes [31], and participation rate in health checks [32]. These factors are also known to be similarly related to the social capital of the regions [33,34,35]. Social capital is the network of relationships in a society based on trust in others, trust in the region they live in, and social participation. Ref. [36] A previous study in Japan showed that a low level of social capital in the region is associated with a high unemployment rate [37]. Considering these studies, a high unemployment rate in the region may badly affect the level of social capital, and that may lead to adverse effects on pregnant women’s behavior. To clarify these mechanisms, further studies about the relation between social capital and smoking habit among pregnant women should be conducted.

A high number of nursery is also related to a high maternal smoking rate in Table 4. In areas with fewer children and aging populations, the number of nursery per child population is especially greater than that in other regions in Japan (Ministry of Health, Labour and Welfare Survey in Japan; 2017). Existing few childrearing events and the loneliness resulting from the small number of women around the same age may increase maternal smoking rate. However, considering the relatively high correlation between the number of nurseries with FSI and population density (as seen in Figure 2), the relationship between the number of nurseries and maternal smoking may have been influenced by these confounding factors. The relationship between maternal smoking and the FSI or population density may have also been obscured by these confounding factors. Residual confounding may have persisted on multiple adjusted levels, as the random parameter in model 4 was not near zero; this may be adjusted with the education level, which we could not include in this study.

In view of the above findings, in areas where the unemployment rate is high or the number of children is very low, the development of a child-rearing environment may play an important role in preventing maternal smoking; this includes encouraging participation in child-rearing events and increasing opportunities for consultation, as these are also related to maternal smoking. In addition, financial or gratuitous support for childcare, and an environment where mothers can raise their children while working should be promoted throughout the region; this will ensure that pregnant women with poor economic status may also raise children properly. Only a few studies have shown the relation between maternal smoking during pregnancy and the characteristic of their living area [10]. In this study, we evaluated this association using nationwide data with an adequate number of pregnant women and using questionnaire surveys administered across 420 regions in the country. We also used sociodemographic data from the statistics bureau in Japan. Further, we used multilevel regression model to investigate the independent effect of individual factors and regional factors on maternal smoking during pregnancy. The validity of this model was improved by selecting variables from several factors with consideration of multicollinearity and Akaike’s Information Criterion.

However, this study also has some limitations. First, we could not identify the number of cigarettes smoked in a day and in which pregnancy period they had smoked. Light and heavy smokers may have been analyzed similarly. Second, it was difficult to validate the self-reported smoking information. Pregnant women may conceal their smoking habits [38,39] or may respond in the negative if they have smoked occasionally during pregnancy. Therefore, the prevalence of smoking among pregnant women may possibly be underestimated. However, the number of pregnant women who provided an incorrect response remains unclear. Third, there might have been regional factors that were not included in the analysis because national data are limited. A previous study has shown that the women’s educational level affects the smoking behavior during pregnancy [19], but we could not use regional variables that reflect the educational level. In addition, regional SES might not have been investigated adequately. We included FSI in the regional variables, but FSI is strongly affected by the wealth of the municipality itself. Other indicators that directly reflect the residents’ financial status such as average income should have been used.

Given that we used only Japanese data in this study, the results may not be generalizable to other countries. Further, we could not show the causal relationship between maternal smoking behavior and the factors found in this study. Further studies are needed to identify these relations by including variables that could not be used in this study, such as the educational level, SES, and social capital.

## 5. Conclusions

The existence of consultation partners, participation in local events for childrearing, and other six factors were significantly associated with maternal smoking behavior during pregnancy. At regional level, unemployment rate and number of nursery were significantly associated with those behaviors after adjusting with individual and regional factors. In areas where the unemployment rate is high or where the number of children is low, the development of a child-rearing environment and improvement of financial or gratuitous support for childcare may help prevent maternal smoking.

## Figures and Tables

**Figure 1 ijerph-17-00173-f001:**
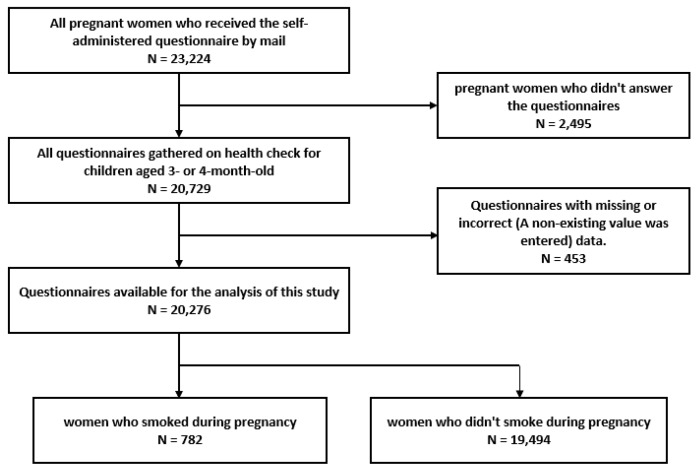
Selection flow of smoking pregnant women.

**Figure 2 ijerph-17-00173-f002:**
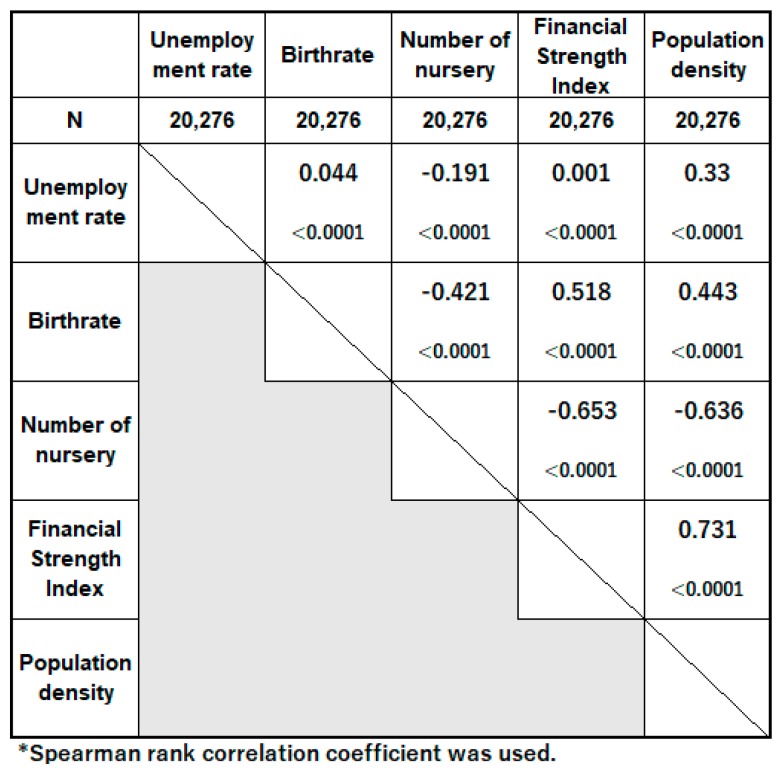
Coefficient of correlation between sociological variables.

**Table 1 ijerph-17-00173-t001:** Summary of Variable Definitions.

Variable Name	Survey Questions, Coding, and Methods of Calculation
Maternal smoke	“Were you smoking during pregnancy?” (1 = No, 2= Yes).
First level variables (N = 20276)	
Birth order	“How many children do you have?” (1 = one, 2= two, 3 = three, 4 = four or more).
Mother’s age at birth	“How old were you at the time of your childbirth?” (1 = 19 years old or younger, 2 = 20 to 24 years old, 3 = 25 to 29 years old, 4 = 30 to 34 years old, 5 = 35 to 39 years old, 6 = 40 years old or older).
Birth weight	“How much did your child’s weight at birth?” (1 = less than 2500 g, 2 = 2500 to 4000 g, 3 = 4000 g or more).
Economic status (subjective)	“How do you feel about your current economic status?” (1 = Very good, 2 = good, 3 = normal, 4 = poor, 5 = very poor).
Husband’s smoking habit during pregnancy	“Was your husband smoking during your pregnancy?” (1 = No, 2 = Yes).
Drinking habit during pregnancy	“Were you drinking during pregnancy?” (1 = No, 2 = Yes).
Existance of counselor	“Are there any counselor during and after your pregnancy?” (1 = No, 2 = Yes).
Participation in local events for child-raising	“Have you participate in local events for child-raising?” (1 = No, 2 = Yes).
Second level variables (N = 420)	
Unemployment rate	(number of unemployed people in the region)/(working population in the region)
Birthrate	annual number of birthchildren per 1000 population in the region.
Number of nursery	number of nurseries per 1000 children population (under five years-old) in the region.
Financial Strength Index(FSI) *	(standard financial revenues in the region)/(amount of basic fiscal demand in the region)
Population density	(population of the region)/(extent of inhabitable area in the region (km^2^))

Data source: Healthy Parents and Children 21, 2013; Ministry of Health, Labour and Welfare, Japan (First level) e-stat, 2010; Statistics Bureau, Ministry of Internal Affairs and Communications, Japan (Second level) * FSI is used as a indicator of municipality’s economic strength in Japan. (more and better).

**Table 2 ijerph-17-00173-t002:** Individual and regional variables.

Variable Name	Smoked During Pregnancy	%	Not Smoked during Pregnancy	%	Prevalence (%)	Variable Name	Smoked during Pregnancy	%	Not Smoked during Pregnancy	%	Prevalence (%)
n	782		19494		3.86						
First level (individual)						Second level (sociological)					
Birth order						Unemployment rate					
First	307	39.4	8870	43.8	3.35	Q1 (0.95–4.75%)	160	20.5	5206	26.7	2.98
Second	230	29.5	7228	35.7	3.08	Q2 (4.75–5.57%)	182	23.3	4825	24.8	3.63
Third	145	18.6	2732	13.5	5.04	Q3 (5.57–6.42%)	205	26.2	4806	24.7	4.09
Fourth or more	97	12.5	625	6.9	13.43	Q4 (6.42–17.5%)	235	30.1	4657	23.9	4.8
Mother’s age at birth						Birthrate					
under 19	22	2.8	206	1.1	9.65	Q1 (1.41–7.39)	244	31.2	5178	25.5	4.5
age 20–24	168	21.5	1850	9.5	8.33	Q2 (7.39–8.15)	266	28.9	4798	23.7	4.5
age 25–29	209	26.7	5676	29.1	3.55	Q3 (8.15–9.26)	134	17.1	4821	23.8	2.7
age 30–34	211	27.0	6792	34.9	3.01	Q4 (9.26–14.1)	178	22.7	4697	23.2	3.65
age 35–39	136	17.4	4183	21.5	3.15	Number of nursery					
over 40	36	4.6	769	4.0	4.47	Q1 (0.00–0.98)	184	23.5	5279	27.1	3.37
Birth weight						Q2 (0.98–1.40)	187	23.9	4740	24.3	3.8
under 2500 g	164	21.0	2589	13.3	5.96	Q3 (1.40–2.04)	185	23.7	4816	24.7	3.7
2500–4000 g	616	78.8	16741	85.9	3.55	Q4 (2.04–7.58)	226	28.9	4659	23.9	4.63
over 4000 g	2	0.3	164	0.8	1.2	Financial Strength Index					
Economic status (subjective)						Q1 (0.11–0.49)	254	32.5	5654	27.9	4.3
Very Good	18	2.3	652	3.4	2.69	Q2 (0.49–0.66)	193	24.7	4538	22.4	4.08
Good	42	5.4	1904	9.8	2.16	Q3 (0.66–0.83)	204	26.1	5012	24.7	3.91
Normal	304	39.2	10804	55.7	2.74	Q4 (0.83–1.72)	131	16.8	4290	21.2	2.96
Poor	253	32.7	4880	25.1	4.93	Population density					
Very Poor	158	20.4	1173	6.0	11.87	Q1 (14–573)	235	30.1	5239	26.9	4.29
Husband’s smoking habit during pregnancy						Q2 (573–1141)	190	24.3	4751	24.4	3.85
Yes	639	86.5	7992	41.3	7.4	Q3 (1141–2652)	180	23	4725	24.2	3.67
No	100	13.5	11374	58.7	0.87	Q4 (2652–1,9260)	177	22.6	4779	24.5	3.57
Drinking habit during pregnancy											
Yes	106	13.6	589	3.0	15.25						
No	672	86.4	18836	97.0	3.44						
Existance of counselor											
Yes	738	95.0	19048	98.0	3.73						
No	39	5.0	381	2.0	9.29						
Participation in local events for child-raising											
Yes	51	6.7	4358	23.0	1.16						
No	706	93.3	14626	77.0	4.6						

**Table 3 ijerph-17-00173-t003:** Single logistic regression analyses.

Variables	OR	95% CI		Variables	OR	95% CI	
First level (individual)				Second level (sociological)			
Birth order				Unemployment rate			
First	ref.			Q1 (0.95–4.75%)	ref.		
Second	0.919	0.773	1.094	Q2 (4.75–5.57%)	1.227	0.989	1.523
Third	1.533	1.253	1.877	Q3 (5.57–6.42%)	1.388	1.125	1.713
Fourth or more	4.485	3.52	5.714	Q4 (6.42–17.5%)	1.642	1.338	2.015
Mother’s age at birth				Birthrate			
under 19	3.438	2.169	5.448	Q1 (1.41–7.39)	ref.		
age 20–24	2.923	2.372	3.603	Q2 (7.39–8.15)	1	0.831	1.203
age 25–29	1.185	0.976	1.44	Q3 (8.15–9.26)	0.59	0.476	0.731
age 30–34	ref.			Q4 (9.26–14.1)	0.804	0.66	0.98
age 35–39	1.047	0.841	1.303	Number of nursery			
over 40	1.507	1.05	2.163	Q1 (0.00–0.98)	ref.		
Birth weight				Q2 (0.98–1.40)	1.132	0.92	1.392
under 2500 g	1.722	1.442	2.055	Q3 (1.40–2.04)	1.102	0.895	1.357
2500–4000 g	ref.			Q4 (2.04–7.58)	1.392	1.141	1.698
over 4000 g	0.331	0.082	1.34	Financial Strength Index			
Economic status (subjective)				Q1 (0.11–0.49)	ref.		
Very Good	0.981	0.606	1.589	Q2 (0.49–0.66)	0.947	0.782	1.146
Good	0.784	0.566	1.086	Q3 (0.66–0.83)	0.906	0.751	1.094
Normal	ref.			Q4 (0.8–1.72)	0.68	0.549	0.843
Poor	1.843	1.554	2.184	Population density			
Very Poor	4.788	3.914	5.856	Q1 (14–573)	ref.		
Husband’s smoking habit during pregnancy				Q2 (573–1141)	0.892	0.733	1.084
Yes	9.093	7.351	11.248	Q3 (1141–2652)	0.849	0.697	1.035
No	ref.			Q4 (2652–1,9260)	0.826	0.677	1.007
Drinking habit during pregnancy							
Yes	1.605	1.373	1.876				
No	ref.						
Existance of counselor							
Yes	ref.						
No	2.313	1.684	3.176				
Participation in local events for child-raising							
Yes	ref.						
No	4.125	3.098	5.491				

**Table 4 ijerph-17-00173-t004:** Multilevel logistic regression analyses (Model 1–Model 4).

**Variables**	**Model 1**	**Model 2**	**Model 3-1**	**Model 3-2**
**OR**	**95% CI**	**OR**	**95% CI**	**OR**	**95% CI**
First level (individual)										
Birth order β1										
First		ref.			ref.			ref.		
Second		1.023	0.842	1.243	1.026	0.845	1.247	1.023	0.842	1.243
Third		1.506	1.187	1.911	1.508	1.189	1.912	1.507	1.188	1.911
Fourth or more		3.104	2.302	4.186	3.076	2.284	4.142	3.068	2.277	4.135
Mother’s age at birth β2										
under 19		3.465	2.063	5.82	3.538	2.109	5.937	3.478	2.072	5.837
age 20–24		2.351	1.84	3.003	2.354	1.843	3.006	2.339	1.831	2.988
age 25–29		1.156	0.933	1.433	1.164	0.94	1.443	1.155	0.932	1.432
age 30–34		ref.			ref.			ref.		
age 35–39		0.981	0.768	1.252	0.992	0.777	1.267	0.978	0.766	1.249
over 40		1.287	0.847	1.953	1.263	0.832	1.918	1.309	0.863	1.985
Birth weight β3										
under 2500 g		1.611	1.319	1.969	1.614	1.321	1.972	1.608	1.316	1.964
2500–4000 g		ref.			ref.			ref.		
over 4000 g		0.202	0.029	1.382	0.199	0.029	1.372	0.21	0.031	1.433
Economic status (subjective) β4										
Very Good		0.754	0.424	1.343	0.75	0.421	1.334	0.761	0.428	1.354
Good		0.872	0.613	1.242	0.887	0.623	1.262	0.869	0.611	1.238
Normal		ref.			ref.			ref.		
Poor		1.434	1.191	1.727	1.436	1.192	1.729	1.438	1.194	1.731
Very Poor		2.462	1.947	3.114	2.473	1.957	3.126	2.465	1.949	3.116
Husband’s smoking habit during pregnancy β5										
Yes		7.361	5.882	9.213	7.367	5.887	9.219	7.333	5.859	9.178
No		ref.			ref.			ref.		
Drinking habit during pregnancy β6										
Yes		3.997	3.066	5.21	3.988	3.063	5.193	3.937	3.021	5.132
No		ref.			ref.			ref.		
Existence of counselor β7										
Yes		ref.			ref.			ref.		
No		2.082	1.411	3.073	2.048	1.39	3.017	2.057	1.394	3.034
Participation in local events for pregnant women β8									
Yes		ref.			ref.			ref.		
No		3.232	2.368	4.411	3.143	2.303	4.29	3.236	2.371	4.415
Second level (sociological)										
Unemployment rate γ01										
Q1 (0.95–4.75%)					ref.					
Q2 (4.75–5.57%)					1.062	0.822	1.373			
Q3 (5.57–6.42%)					1.41	1.102	1.805			
Q4 (6.42–17.5%)					1.477	1.16	1.88			
Birthrate γ02										
Q1 (1.41–7.39)								ref.		
Q2 (7.39–8.15)								1.079	0.861	1.354
Q3 (8.15–9.26)								0.714	0.556	0.917
Q4 (9.26–14.1)								0.972	0.767	1.233
Fixed effect (level1)										
Intercept γ00	−3.28		−6.41			−6.59			−6.34	
Random parameter (level2)										
Between community u0j	0.158		0.092			0.053			0.065	
AIC	6608		4981			4972			4976	
**Variables**	**Model 3-3**	**Model 3-4**	**Model 3-5**	**Model 4**
**OR**	**95% CI**	**OR**	**95% CI**	**OR**	**95% CI**	**OR**	**95% CI**
First level (individual)												
Birth order β1												
First	ref.			ref.			ref.			ref.		
Second	1.017	0.837	1.236	1.021	0.84	1.24	1.025	0.843	1.245	1.022	0.841	1.241
Third	1.49	1.174	1.891	1.501	1.182	1.905	1.512	1.191	1.92	1.489	1.174	1.889
Fourth or more	3.045	2.256	4.109	3.102	2.299	4.186	3.124	2.315	4.216	2.997	2.223	4.04
Mother’s age at birth β2												
under 19	3.46	2.06	5.809	3.457	2.058	5.807	3.483	2.073	5.851	3.602	2.148	6.039
age 20–24	2.335	1.828	2.984	2.354	1.843	3.008	2.359	1.847	3.015	2.338	1.831	2.985
age 25–29	1.152	0.929	1.428	1.157	0.933	1.434	1.159	0.935	1.437	1.164	0.939	1.442
age 30–34	ref.			ref.			ref.			ref.		
age 35–39	0.979	0.767	1.251	0.983	0.77	1.255	0.98	0.768	1.251	0.992	0.777	1.266
over 40	1.293	0.851	1.964	1.289	0.848	1.958	1.278	0.841	1.941	1.283	0.845	1.949
Birth weight β3												
under 2500 g	1.608	1.315	1.964	1.612	1.319	1.97	1.615	1.322	1.974	1.609	1.318	1.965
2500–4000 g	ref.			ref.			ref.			ref.		
over 4000 g	0.192	0.026	1.391	0.205	0.03	1.41	0.201	0.029	1.38	0.212	0.031	1.47
Economic status (subjective) β4												
Very Good	0.752	0.422	1.338	0.754	0.423	1.342	0.754	0.424	1.342	0.747	0.419	1.329
Good	0.87	0.611	1.239	0.873	0.613	1.243	0.872	0.612	1.241	0.883	0.62	1.256
Normal	ref.			ref.			ref.			ref.		
Poor	1.433	1.19	1.726	1.433	1.189	1.725	1.433	1.19	1.726	1.443	1.199	1.737
Very Poor	2.472	1.954	3.127	2.456	1.941	3.107	2.461	1.946	3.112	2.487	1.968	3.142
Husband’s smoking habit during pregnancy β5												
Yes	7.332	5.858	9.177	7.347	5.869	9.197	7.386	5.9	9.246	7.311	5.842	9.148
No	ref.			ref.			ref.			ref.		
Drinking habit during pregnancy β6												
Yes	4.011	3.075	5.231	3.995	3.064	5.208	3.979	3.051	5.189	3.95	3.032	5.147
No	ref.			ref.			ref.			ref.		
Existence of counselor β7												
Yes	ref.			ref.			ref.			ref.		
No	2.093	1.418	3.091	2.075	1.405	3.063	2.077	1.408	3.064	2.027	1.375	2.987
Participation in local events for pregnant women β8											
Yes	ref.			ref.			ref.			ref.		
No	3.242	2.375	4.424	3.224	2.362	4.4	3.222	2.361	4.397	3.165	2.319	4.319
Second level (sociological)												
Unemployment rate γ01												
Q1 (0.95–4.75%)										ref.		
Q2 (4.75–5.57%)										1.001	0.779	1.287
Q3 (5.57–6.42%)										1.439	1.118	1.852
Q4 (6.42–17.5%)										1.481	1.147	1.914
Birthrate γ02												
Q1 (1.41–7.39)										ref.		
Q2 (7.39–8.15)										0.967	0.763	1.226
Q3 (8.15–9.26)										0.638	0.488	0.834
Q4 (9.26–14.1)										0.915	0.696	1.203
Number of nursery γ03												
Q1 (0.00–0.98)	ref.									ref.		
Q2 (0.98–1.40)	1.112	0.861	1.435							1.134	0.893	1.441
Q3 (1.40–2.04)	0.967	0.746	1.252							1.068	0.812	1.404
Q4 (2.04–7.58)	1.185	0.927	1.517							1.406	1.044	1.894
Financial Strength Index γ04												
Q1 (0.11–0.49)				ref.						ref.		
Q2 (0.49–0.66)				1.039	0.82	1.318				1.155	0.882	1.511
Q3 (0.66–0.83)				1.122	0.889	1.417				1.311	0.977	1.76
Q4 (0.83–1.72)				0.899	0.688	1.173				1.151	0.805	1.646
Population density γ05												
Q1 (14–573)							ref.			ref.		
Q2 (573–1141)							1.051	0.828	1.334	0.968	0.744	1.258
Q3 (1141–2652)							1.053	0.824	1.346	0.984	0.71	1.363
Q4 (2652–19,260)							1.094	0.852	1.405	1.032	0.733	1.452
Fixed effect (level1)												
Intercept γ00		−6.472			−6.431			−6.461			−6.700	
Random parameter (level2)												
Between community u0j		0.089			0.092			0.089			0.037	
AIC		4983			4985			4987			4974

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
