# Peer review of "Individual and Regional Characteristics Associated with Maternal Smoking During Pregnancy in Japan: Healthy Parents and Children 21"

_ijerph, 2019, doi:10.3390/ijerph17010173_

Round 1

Reviewer 1 Report

Thank you for the opportunity to review this manuscript examining the association between maternal smoking during pregnancy and individual and regional characteristics. This is an important area of research.

The introduction to this manuscript could be strengthened with a more thorough literature review. It could also benefit with an explanation as to why the authors had good reason to believe that the individual and regional characteristics they chose to examine play a role in smoking during pregnancy in Japan. In the other hand the conclusion should focus more on how the results of this study could guide decision makers to reduce the smoking rates among pregnant women in Japan. What measures do the authors propose?

A limitation of this study is how the authors assessed smoking; although they mention in the limitations I believe the implications go further than what they mentioned. Some people who smoke occasionally will answer no to a single item question; thus several pregnant women who were categorized as non-smokers might have smoked.  

A few more minor points:

Several references are missing throughout the manuscript(e.g. sentence in line 180; line 184). An explanation of what is meant by “incorrect data” in Figure 1 is needed.

Reviewer 2 Report

The work presented in this manuscript is on the personal and regional factors associated with maternal smoking during pregnancy. The sample size is sufficient for this kind of study and data analyses are seemingly appropriate. The findings are not so novel but reasonable. Basically, this manuscript is suitable for publication in IJERPH.

General comment

The manuscript is written by readable English but it still requires extensive editing. Discussion on the individual factors significantly associated with maternal smoking is required. Probably, the authors did not discuss them because these were already reported as factors significantly associated with maternal smoking in previous studies; however, the literature findings were mostly from populations in western countries where culture and life style must be different from Japan. The fact that the factors are significantly related to maternal smoking under different settings may be worth discussing. Unfortunately, the authors’ inference on the association between maternal smoking and regional factors were not convincing. Considering the highly significant inter-correlations between regional factors shown in Fig. 2, the authors may need to discuss possible confounding effects that could not be fully corrected for in this multivariate analysis

The following points should be clarified.

Page 2, line 64. What administrative unit of this country was assigned for the “municipality” in this study? Page 3, line 77. Clearly state to what were the 18 questions significantly related. Page 3, line 83. Clearly state with what were the 5 regional factors significantly correlated. 2 and Fig. A1. Specify N of the correlation analyses. Also specify what correlation analytical method was used. Page 9, line 155-156. The result on variation in maternal smoking among regions were not given in RESULT section. Page 10, line 206. Is it the authors’ mistake to mention about educational levels and SES as regional factors? These are clearly individual factors.

Reviewer 3 Report

Very interesting article.

Please explain in the introduction, the association of variables (individual and regional), specially the interaction that authors are expected to find. Causal association behind the analysis, is not completely clear.

Please give more antecedents about model construction. Specifically, why authors included those selected variables. 

Figure 2 does not improve the understanding of the association between different variables. Table 3 is a better way to understand the strength of association, more that coefficient of correlation, I suggest to eliminate figure 2.

Round 2

Reviewer 2 Report

The authors have made appripriate revisions to the manuscript.